# Blastospores from *Metarhizium anisopliae* and *Metarhizium rileyi* Are Not Always as Virulent as Conidia Are towards *Spodoptera frugiperda* Caterpillars and Use Different Infection Mechanisms

**DOI:** 10.3390/microorganisms11061594

**Published:** 2023-06-16

**Authors:** Isabella Alice Gotti, Camila Costa Moreira, Italo Delalibera, Henrik H. De Fine Licht

**Affiliations:** 1Department of Entomology and Acarology, “Luiz de Queiroz” College of Agriculture, University of São Paulo, Piracicaba 13418-260, Brazil; delalibera@usp.br; 2R&D Microbiologicals Department, Koppert Biological Systems Brazil, Piracicaba 13400-970, Brazil; camilabio7@gmail.com; 3Section for Organismal Biology, Department of Plant and Environmental Sciences, University of Copenhagen, 1165 Copenhagen, Denmark; hhdefinelicht@plen.ku.dk

**Keywords:** blastospores, conidia, cuticle penetration, liquid fermentation, transcriptome, virulence

## Abstract

Infective conidia from entomopathogenic fungi are widely used to control insect pests. Many entomopathogenic fungi also produce yeast-like cells called blastospores under specific liquid culture conditions that can directly infect insects. However, little is known about the biological and genetic factors that allow blastospores to infect insects and make them potentially effective for biological control in the field. Here, we show that while the generalist *Metarhizium anisopliae* produces a higher number of and smaller blastospores, the Lepidoptera specialist *M. rileyi* produces fewer propagules with a higher cell volume under high-osmolarity conditions. We compared the virulence of blastospores and conidia of these two *Metarhizium* species towards the economically important caterpillar pest *Spodoptera frugiperda*. Conidia and blastospores from *M. anisopliae* were equally infectious, but acted slower, and killed fewer insects than *M. rileyi* conidia and blastospores did, where *M. rielyi* conidia had the highest virulence. Using comparative transcriptomics during propagule penetration of insect cuticles, we show that *M. rileyi* blastospores express more virulence-related genes towards *S. frugiperda* than do *M. anisopliae* blastospores. In contrast, conidia of both fungi express more virulence-related oxidative stress factors than blastospores. Our results highlight that blastospores use a different virulence mechanism than conidia use, which may be explored in new biological control strategies.

## 1. Introduction

Biological control agents are one of the pillars of integrated pest management (IPM), which is a promising tool for reducing the impact of pests. Microbial pathogens, such as insect pathogens, are a valuable tool to manage crop pests and can be used in combination with insecticides to improve their performance and reduce the occurrence of resistance to commercially active ingredients [1].

Fungi are common pathogens of insects and natural regulators of pest insect populations [2]. A notable example is fungi from the genus *Metarhizium* (Ascomycota: Hypocreales), which are employed for the biological control of crop pests and vector-borne diseases [2,3]. Several *Metarhizium* species have multifunctional lifestyles and occur in several ecological niches, including as insect pathogens, plant root symbionts, and soil saprophytes [4]. The species *M. anisopliae* (Metsch.) Sorokin, which in Brazil is used to control sugarcane pests, is one of the best examples of using an entomopathogenic fungus as pest control [5].

Entomopathogenic fungi are produced primarily in vitro on a solid medium or under liquid fermentation to obtain different structures such as conidia, blastospores, or mycelium. Except for conidia, the blastospores and mycelium are produced using liquid culture media [6]. The natural life cycle of entomopathogenic fungi involves different fungal structures and cell types. Fungal conidia ensure dissemination in the environment and transmission to new insect, plant, and soil niches, whereas blastospores are naturally formed when these fungi proliferate inside the host hemolymph. Blastospores thus naturally differ from conidia in several ways because they are hydrophilic, germinate faster than conidia (2–8 h versus 12–24 h), and are often more virulent towards susceptible hosts [7]. The rapid germination rate of blastospores produced under liquid culture conditions could be regarded as a virulence determinant and is a desirable trait for use as an infective structure in commercially produced entomopathogens [8,9].

Much progress has been made in production and formulation technologies, which coupled with knowledge concerning the target insect’s biology, make entomopathogenic fungi attractive components of integrated pest management practices [10]. It is possible to produce large amounts of blastospores in a small space by liquid fermentation in a short time (<4 days) [11,12,13,14]. Furthermore, compared with solid fermentation methods, liquid fermentation requires less manual labor and does not rely on a high-value substrate such as rice grains making liquid fermentation potentially more cost effective in a commercial production setting [12].

The blastospores of *Beauveria bassiana* (Balsamo) Vuillemin (Ascomycota: Hypocreales) produced in high-osmolarity liquid fermentation are smaller and more virulent towards pest insects than are blastospores produced without osmotic stress [15]. A better understanding of the biological and genetic factors behind blastospore production and virulence could help enable this propagule to be used as an active ingredient of commercially produced biological control products.

Comparative genomics has facilitated identifying fungal fitness traits and the selective forces that act upon them to improve our understanding of how and why entomopathogenic fungi interact with insects and other components of their environments [16]. These technologies have helped to identify the functions of several pathogenicity genes of entomopathogenic fungi [3,6,17]. However, in-depth studies of germination and infection of blastospores and conidia in insects are needed to better understand these fungi and expand their biotechnological potential. Genome studies of entomopathogenic fungi have helped us understand the potential possessed by these fungi both as insect pathogens and as microbial biocatalysts.

The fungus *Metarhizium rileyi* (Farlow) Kepler, S.A. Rehner and Humber, previously known as *Nomuraea rileyi*, is pathogenic to Lepidoptera, infecting noctuids such as *Anticarsia gemmatalis* and *Spodoptera frugiperda*, which are critical pests of important crops [18,19]. It is a dimorphic fungus with yeast-like hyphal bodies and a filamentous growth phase [18]. Much less is known about *M. rileyi* compared to *M. anisopliae*, but recent advances in *M. rileyi* research have contributed to our knowledge of different aspects of this pathogen, and this species’ potential as an entomopathogen is indisputable [18,19,20,21].

*Spodoptera frugiperda* (J.E. Smith, 1797), known in English as the fall armyworm, is a polyphagous pest reported to infest 353 host plant species in North and Central America [22]. It prefers wild and cultivated grasses, maize, rice, sorghum, millet and sugarcane, cultures that are overspread in Brazil, reinforcing the importance of finding new strategies of its control.

Here, we first compare the virulence of *M. anisopliae* and *M. rileyi* conidia and blastospores towards *Spodoptera frugiperda* larvae. Second, we use comparative transcriptome analysis of *M. anisopliae* and *M. rileyi* during infection of *S. frugiperda* cuticles to identify virulence-related genes expressed specifically during cuticle penetration. We intend to compare the virulence of *M. anisopliae* and *M. rileyi* spores towards *S. frugiperda* larvae. Additionally, we characterize the gene expression profiles of fungal propagules grown on caterpillars’ cuticles. These findings can help us to understand the forces that act upon entomopathogen fitness traits and may underpin both our understanding of the natural roles of these fungi in nature and the further development of new mycoinsecticides.

## 2. Materials and Methods

### 2.1. Fungal Isolates and Insect Rearing

Two isolates of *Metarhizium anisopliae* (ESALQE9 and ESALQ4676) and two isolates of *Metarhizium rileyi* (ESALQ4946 and ESALQ5611) were selected from the entomopathogen collection “Prof. Sergio Batista Alves” of the Insect Pathology and Microbial Control Laboratory, University of São Paulo, Piracicaba, Brazil (ESALQ/USP). The fungi were cultured in Petri dishes containing potato dextrose agar (PDA) for *M. anisopliae*, and Sabouraud maltose agar, supplemented with yeast extract (modified SMAY) for *M. rileyi*. Fungal cultures were incubated in BOD (biological oxygen demand) at 26 °C for ten days before harvesting the conidia for the assays.

The *Spodoptera frugiperda* eggs were purchased from specialized breeders and kept under consistent laboratory conditions (temperature, 27 ± 1 °C; photoperiod, 14:10 L:D; relative humidity, 70 ± 10%). Hatched caterpillars were transferred to plastic pots, and fed on corn leaves (*Zea mays*) until the second instar (4–5 days after the hatching of the eggs).

### 2.2. Blastospore and Conidium Production

Suspensions of *M. anisopliae* and *M. rileyi* conidia were obtained by washing sporulated fungus plates with 10 mL of a 0.01% aqueous solution of Tween^®^ 80. These suspensions were used to inoculate a pre-culture liquid medium in the standard concentration of 5 × 10^6^ conidia/mL for *M. anisopliae*, and 5 × 10^7^ conidia/mL for *M. rileyi* production. The liquid culture medium was a basal medium supplemented with 45 g/L of yeast extract and 80 g/L or 200 g/L of a carbon source to create osmotic stress. The carbon source (20% *w*/*v*) was autoclaved separately from the salt solution and added before the fungus inoculation. The basal media contained, per liter of distilled water, 2.0 g of KH_2_PO_4_, 0.4 g of CaCl_2_, 0.3 g of MgSO_4_, 0.05 g of FeSO_4_, 37 mg of CoCl_2_, 16 mg of MnSO_4_, 14 mg of ZnSO_4_, 500 μg of Thiamine, Riboflavin, Pantothenate, Niacin, Pyridoxamine, and Thioctic Acid each, and 50 μg of Folic Acid, Biotin, and Vitamin B12 [23,24].

Fungal cultures were grown in 250 mL “baffled flasks” with 50 mL of the medium containing the fungal inoculum in each flask. Cultures were kept in an orbital shaker (350 rpm, 28 °C) and shaken daily by hand to minimize mycelial growth on the flask’s wall. The concentration of 80 g of the carbon source per L^−1^ was considered the standard treatment. Two shake flasks (replicates) per experimental treatment were used for each experiment, and all experiments were independently repeated three times (*n* = 6). Blastospore concentration was determined microscopically using a Neubauer chamber.

To determine the effect of the osmotic stress on cell size and morphology, the length and width of 50 random blastospores from each treatment flask were measured using a Leica Biosystems light microscope (Wetzlar, Germany) and image system (Leica DM4B; LAS Version 4.1.0) at a 400× magnification. Subsequently, the cell size was indirectly inferred by the volume in cubic micrometers (µm^3^) of an ellipsoid given by Equation (1), where *a* is the length and *b* is the width [15].
(1)$$V\;\left({\mu m}^{3}\right)=\frac{4}{3}\times \left(\pi \times a\times b^{2}\right)$$

Fungal conidia were produced in Petri dishes (90 × 10 mm) with a solid medium according to the fungi species. The PDA medium was used for *M. anisopliae* and the modified SMAY medium was used for *M. rileyi*. The plates were incubated for 15 days at 27 ± 1 °C with a relative humidity (RH) higher than 90%.

The mean blastospore yield produced by each fungus in each osmotic level was analyzed by generalized linear models (GLM) with quasipoisson distribution and compared via analysis of variance (ANOVA) with a chi-square test. Pairwise comparisons between treatments were carried out using the general linear hypothesis (function glht) in the multcomp R package [25]. Data overdispersion was checked by the half-normal plots (function hnp) implemented at hnp R package [26]. The blastospore volumes obtained by each fungus in both osmotic levels were analyzed using linear models with normal distribution compared via ANOVA with an F test followed by Tukey’s HSD post hoc test with a *p* <  0.05 level of probability.

### 2.3. Virulence Bioassay

Blastospores produced in liquid media under different osmotic stress conditions and conidia produced in solid media were used for the virulence tests. To avoid the influence of possible metabolites dispersed in the medium, after the development of the fungus in its respective liquid culture media, blastospores were separated from the media before spray application on the *S. frugiperda* caterpillars. Blastospores were collected via filtration through 2 layers of lens cleaning cloth (Whatman n° 105) and washed twice with a potassium buffer saline solution (composition per liter: NaCl, 8.0 g; KCl, 0.2 g; Na_2_HPO_4_, 1.44 g; KH_2_PO_4_, 0.24 g. pH adjusted to 6.0). During this process, the medium was centrifuged at 5000 rpm for 10 min at each wash. Fresh conidia were harvested with a microbiological loop and suspended in 10 mL of 0.01% (*v*/*v*) Tween^®^ 80. All suspensions were vortexed to ensure the samples’ homogeneity, and concentrations were determined microscopically using a Neubauer chamber. The suspensions were adjusted to a concentration of 5 × 10^7^ propagules/mL dose, for blastospores and for conidia.

Using a potter spray tower (Burkard Manufacturing Co., Ltd., Rickmansworth, UK), 90 mm Petri dishes containing 16 second-instar caterpillars received a volume of 2 mL of a fungal suspension for each treatment. The treatments consisted of fresh blastospores of two different carbon source levels, fresh aerial conidia, and a control group that received only a 0.05% aqueous solution of Tween^®^ 80. After 24 h, the caterpillars were placed individually into wells of plastic trays, and they were fed with corn leaves during the evaluation period. Caterpillar mortalities were evaluated daily until day 7. Dead larvae were surface-disinfected by soaking them in 70% ethanol, transferred to humid chambers and kept at 28 °C to confirm if the mortality was caused by the fungus. Confirmed mortality was analyzed by visualizing morphologically the fungal sporulation.

The second-instar caterpillars were used to test the virulence of fungi for two reasons: (1) first-instar caterpillars are highly susceptible and naturally experience high mortality rates [27]; (2) in practical terms, caterpillars larger than the third instar are less susceptible and have a habit of seeking refuge in the whorls of corn plants, where the probability of fungal infection decreases [28].

Survival analysis was performed with censored data for dead larvae until day 7 using a parametric model for survival data with a Weibull distribution (survival r-package) [29]. The survival curves were compared by a log–likelihood ratio test at *p* < 0.05. The insect-confirmed mortalities (mycosis) of each fungus were compared between the treatments using generalized linear models (GLM) and quasi-Poisson distribution. Data were analyzed via ANOVA with an F test and pairwise comparisons were carried out using the function glht in the multcomp r-package [25] to compare treatments. Data overdispersion was checked using the function hnp implemented in the hnp r-package [26]. The lethal times (LT50 and LT95) were calculated using a probit analysis in the ecotox r-package [30].

### 2.4. Microscopy of Propagule Germination and Penetration of Insect Cuticle

One isolate of each fungal species was selected for the penetration studies, RNA extraction studies and transcriptomic studies. *M. anisopliae* ESALQ E9 was selected for being a commercial generalist isolate, being a registered product in Brazil for spittlebugs on sugarcane and pasture. The other one, *M. rileyi* ESALQ 5611, was selected for being isolated from *S. frugiperda* caterpillars as a guarantee of its specificity.

Scanning electron microscopy (SEM) was used to determine if fungal propagules would form appressoria structures on insect cuticles and to determine the best time point for RNA extractions. Suspensions of *M. anisopliae* and *M. rileyi* propagules were sprayed on *Spodoptera frugiperda* caterpillars, and different time points after spraying were selected. Petri dishes containing five third-instar caterpillars received a volume of 2 mL of a fungal solution of concentration 1 × 10^7^ propagules mL with a Potter Spray Tower. The treatments were new blastospores produced with 200 g L^−1^ of glucose and new aerial conidia from each *M. anisopliae* and *M. rileyi* isolate. After the caterpillars’ spraying with blastospores and conidia, the samples were collected and prepared for analysis by SEM techniques. Analyzed times were 2 h, 4 h, 6 h, and 8 h after spraying for blastospores and 12 h, 16 h, 24 h, and 32 h after spraying for aerial conidia.

Caterpillars were anesthetized by cold shock (refrigerator at 4 °C for 5 min) and subsequently fixed in 4% paraformaldehyde in phosphate-buffered solution (PBS), for 48 h, at 4 °C. This was followed by dehydration in an increasing series of acetone (70%, 80%, 90%, 95%, and twice in 100%, lasting 10 min for each batch). After drying at an Automated Critical Point Dryer Leica EM CPD300 (Washington, DC, USA), they were glued in aluminum stubs using double-sided tape until the samples were metalized with gold in the Sample Sputter Coater Balzers SCD 050 (Washington, DC, USA). The specimens were examined and photographed using a JEOL JSM-IT300 scanning electron microscope (Akishima, Japan) operated at 15.0 kV.

### 2.5. RNA Extraction and Sequencing

Before RNA extractions, *S. frugiperda* caterpillars were inoculated with *M. anisopliae* (ESALQE9) or *M. rileyi* (ESALQ5611) by soaking them in suspensions of the conidia or blastospores of each isolate. Each replicate consisted of seven 4th-instar caterpillars. Five replicates were used, totalizing 20 samples. Fungal suspensions were prepared at a concentration of 5 × 10^7^ propagules mL for each treatment.

After inoculation, *S. caterpillars* were fed with corn leaves and kept under controlled temperature and humidity conditions for 6 h for blastospore treatments and 24 h for conidium treatments (time at which appressoria are formed according to SEM analysis). After that, the digestive tube of the caterpillars was removed on ice, and the cuticle was used for RNA extraction. The RNA was extracted using TRIZOL^®^ (Invitrogen Life Technologies, Waltham, MA, USA). The protocol was carried out according to the manufacturer’s recommendations [31]. After extraction, the total RNA was treated with RQ1 RNAse-free DNase Promega according to the usage information [32].

Total RNA was quantified in the fluorometer (Qubit, Invitrogen), and the concentration and quality of the samples were evaluated via analysis using the spectrophotometer NanoDrop ND-1000 (Wilmington, NC, USA). The RNA integrity was assessed on 1% agarose–formaldehyde gel. Samples were sequenced with Illumina HiSeq 2500 technology (San Diego, CA, USA), which yielded at least 20 million 150 bp paired-end reads per library. Library preparation and sequencing were performed by NGS Soluções Genômicas in Piracicaba-SP, Brazil.

### 2.6. RNA-Seq Reads

The quality of the raw reads before and after quality and adaptor trimming was assessed using the FastQC program. Illumina adapters and low-quality sequences were removed using TrimGalore v0.6.4, AfterQC v0.9.6, and Trimmomatic v0.32 with the following options: LEADING:20 TRAILING:20 SLIDINGWINDOW:4:20 HEADCROP:7 MINLEN:36 [33]. To compile the FastQC reports and identify patterns between the samples of each treatment, the MultiQC program was used before and after read trimming.

To align reads to the genome, the reference genomes of *M. anisopliae* and *M. rileyi* were downloaded from the National Center for Biotechnology Information (NCBI) search database. The GenBank assembly accession GCA_013305495.1 (JEF-290 strain) was chosen for *M. anisopliae* and the GenBank assembly accession GCA_007866325.1 (Cep018-CH2) was chosen for *M. rileyi*. The annotation files were downloaded as well. Filtered reads were mapped to reference genomes using the HISAT2 software v2.2.1, following standard settings (available at https://github.com/DaehwanKimLab/hisat2 accessed on 10 January 2022). RNA-seq reads are available at the National Library of Medicine (NCBI) with the BioProject accession number: PRJNA976674.

### 2.7. Differential Gene Expression Analysis

Before running the differential gene expression analysis, the gene count matrixes were obtained with the python script provided by John Hopkins University, Center for Computational Biology (available at http://ccb.jhu.edu/software/stringtie/index.shtml?t=manual#deseq accessed on 10 January 2022). The gene count matrixes were used as input files for the differential expression analysis conducted using the DESeq2 package from the statistical software R (R 4.2.2 version).

Before running the DESeq2 package, pre-filtering and sample-to-sample distances were assessed for all treatments using principal component analysis (PCA) plots. Rows with reading counts of <5 were excluded, and genes with adjusted *p*-values of <0.1 were considered differentially expressed. Diagnostic plots (an MA plot and volcano plot) were prepared and analyzed for each treatment. Heat maps of differentially expressed genes were made using the heatmap.2 packages from the R software.

The InterPro database was used to annotate protein sequences and better identify protein domains of the differential expressed genes [34]. *Metarhizium anisopliae* and *Metarhizium rileyi* protein sequences were downloaded from NCBI and used as input in InterProScan [35]. Then, only mRNA was filtered from the output for reference in the gene–protein analysis.

## 3. Results

### 3.1. Blastospore Production under Different Levels of Osmotic Stress

Growth of *M. anisopliae* at a high carbon source concentration of 200 g/L resulted in a higher blastospore yield of both isolates compared to the production of fungus under the standard concentration of 80 g/L (*z =* 2.055, *p =* 0.039917; Figure 1A). A comparison of the two *M. anisopliae* isolates showed that the isolate ESALQE9 responds better to the osmotic stress in liquid fermentation, resulting in higher yields. The blastospore production by the isolate ESALQ 4676 did not differ between the different sugar levels (*z =* −1.363, *p =* 0.1729, Figure 1A). The high carbon source concentration resulted in smaller *M. anisopliae* cells than those produced under the standard concentration for both isolates (*p <* 0.0001). When grown with 80 g of the carbon source per L^−1^, both isolates showed the same pattern with a mean blastospore volume of 1050 µ3 compared to that of 730 µ3 when produced under 200 g of the carbon source per L^−1^ (Figure 1C,D).

*Metarhizium rileyi* blastospore production under different concentrations of the carbon source showed a different pattern under the same osmotic stress conditions. While the *M. anisopliae* total cell volume reduces under osmotic stress conditions during liquid fermentation, that of *M. rileyi* develops reversely. The high carbon source concentration did not increase the blastospore production; on the other hand, blastospore production was higher in a low sugar concentration for the isolate ESALQ 4946 (*z =* 7.456, *p <* 0.001; Figure 1B) and for the isolate ESALQ 5611 (*z =* 12.44, *p <* 0.001). Additionally, the higher carbon source concentration did not result in smaller blastospores of *M. rileyi* (Figure 1B). A clear pattern of blastospore sizes was not observed; however, isolate ESALQ 4676 produced smaller blastopores under a lower carbon source concentration (*p <* 0.001). Blastopores produced by the isolate ESALQ 5611 did not present size differences regardless of the sugar concentration (*p =* 0.2232).

### 3.2. Virulence Bioassay

The *M. anisopliae* isolate (ESALQ E9), a generalist fungus isolated from the soil, acted slowly, and killed less than 50% of the insects in 7 days with all the propagules at a concentration of 5 × 10^7^ mL^−1^. This result is strongly supported by the median lethal time calculated for the propagules (LT50). *M. anisopliae* conidia had an estimated LT50 of 7.5 days, compared to that of blastospores produced under 80 g of the carbon source per L^−1^ of 18.5 days, and 20.6 days for blastospores produced under 200 g of the carbon source per L^−1^ (Table 1).

On the other hand, *M. rileyi* (ESALQ 5611), a specialist fungus isolated from a *Spodoptera frugiperda* caterpillar, killed 50% of the evaluated insects within 3–6 days, with all the propagules at a concentration of 5 × 10^7^ mL^−1^. Conidia attained the highest LT50 value (2.57 days), while the lowest LT50 value was attributed to blastospores produced under 80 g of the carbon source per L^−1^ (6.28 days). Comparing blastospore types, the propagules produced under 200 g of the carbon source per L^−1^ were more efficient at killing *S. frugiperda* caterpillars (4.84 days) (Table 2).

The survival of the *S. frugiperda* caterpillars that received the control treatment and that of the *M. anisopliae* propagules were different (*χ^2^ = *25.98, *p <* 0.001). There was no difference in caterpillar mortality between the *M. anisopliae* blastospores and conidia applied to the caterpillars (*p* < 0.83735). Comparing fungal species, *M. anisopliae* propagules were less virulent to *S. frugiperda* caterpillars (the mortality rate was less than 25% for all treatments) and resulted in less than 5% mycosed insects irrespective of the infective propagule (Figure 2B) with a significant difference between the propagules and the control (*F* = 4.5292, *p* = 0.01037). Therefore, despite producing more blastospores and smaller cells, the osmotic stress did not produce more virulent propagules (*p >* 0.583).

The survival of *S. frugiperda* differed between that of the control and the *M. rileyi* propagules (*χ^2^ = *177.46, *p <* 0.001). The conidia of *M. rileyi* were more virulent than both blastospores (*p* < 0.001), even those produced under a high concentration of the carbon source. Blastospores produced using 200 g of the carbon source per L^−1^ were more virulent than those produced on the standard carbon source concentration (*p* < 0.001). Blastospores produced using 200 g of the carbon source per L^−1^ also resulted in more mycosed insects than did the other propagules and the control (Figure 2C,D).

### 3.3. Propagule Germination and Penetration through the Insect Cuticle

Image analysis revealed that blastospores germinate between 2 and 4 h after spraying and penetrate between 6 and 8 h after spraying. The exact penetration was not possible to see. Comparing the images at 6 h, when both the blastospore and germ tube were turgid with the images of the collapsed blastospore at 8 h after spraying, it is possible to infer that the blastospores penetrated sometime between 6 and 8 h after spraying (Figure 3A).

Furthermore, conidia images showed that germination occurred 16 h after spraying and estimated penetration occurred between 24 and 32 h after spraying. A similar phenomenon is observed in blastospores; by 24 h, the conidia and germ tube seemed turgid. At 32 h, the conidia seemed collapsed, letting us infer that those penetrations occurred (Figure 3B).

### 3.4. RNAseq Sequencing and Differential Gene Expression Analysis

Based on an analysis of SEM pictures, replicate samples were collected for RNAseq analysis 24 h post-exposure for *Metarhizium anisopliae* and 8 h post-exposure for *Metarhizium rileyi*. Analyzing the number of mapped reads in each sample, we found that less than 1% of the reads were uniquely mapped to the respective fungi’s genome (Appendix A). When the same samples were mapped with the *Spodoptera frugiperda* reference genome, it was possible to observe almost 70% of uniquely mapped reads, confirming that the low alignment to the fungal genomes was because of a reduced amount of fungal RNA in the samples, compared to the large amount of insect RNA (Appendix A).

The principal component analysis of the *M. anisopliae* and *M. rileyi* propagules during the penetration through *S. frugiperda* caterpillars showed a good separation between blastospores and conidia for both fungi (Figure 4). The PCA plots largely separate treatments and explain 58% of the sample variation in *M. anisopliae* (Figure 4A) and 33% of that in *M. rileyi* (Figure 4B). For *M. anisopliae*, reads were mapped to 4870 genes in the reference genome with 319 genes being differentially expressed between blastospores and conidia (FDR-adjusted *p <* 0.1; Log2 fold change (*FC*) > 2 or <−2). Out of these 319 genes, 36 genes were up-regulated in blastospores, and 283 were up-regulated in conidia (Figure 5A). For *M. rileyi*, reads were mapped to 6128 genes in the reference genome with 101 genes being differentially expressed between blastospores and conidia (FDR-adjusted *p <* 0.1, Log2FC > 2 or <−2). Out of these differentially expressed genes, 39 genes were up-regulated in blastospores, and 62 were up-regulated in conidia (Figure 5B).

Protein domain analysis of differentially expressed genes revealed 194 protein family (PFAM) terms among up-regulated genes in *M. anisopliae* conidia, and 19 of those in *M. anisopliae* blastospores. For *M. rileyi*, we found 22 up-regulated PFAM terms in blastospores and 28 of those in conidia (Figure 5C). Conidia from *M. anisopliae* and *M. rileyi* share 19 PFAM terms related to virulence factors. In contrast, blastospores from both species only share one PFAM term corresponding to a heat shock protein (HSP) (Figure 5C, Appendix A). All the treatments share only one PFAM: cytochrome P450, an oxidative stress factor (Figure 5C, Appendix A).

Up-regulated genes of *M. anisopliae* and *M. rileyi* propagules were grouped via PFAM annotation (Figure 5D). Analyzing the amount of PFAM terms presented in different functions, it was observed that, in general, conidia of both fungi expressed more virulence and oxidative stress factors than did blastospores. *M. rileyi* blastospores expressed more of these factors against *S. frugiperda* than did *M. anisopliae* blastospores. To better analyze the virulence of *M. anisopliae* and *M. rileyi* propagules towards *S. frugiperda* caterpillars, we focused only on PFAMs related to virulence and oxidative stress factors (Figure 6). In this analysis *M. anisopliae* conidia had 72 unique virulence-related elements compared to the 11 virulence-related elements *M. rileyi* had. Conidia from both species share ten genes from all those virulence-related factors, while blastospores share no one.

## 4. Discussion

The present study focused on investigating the impact of glucose concentration on *Metarhizium* blastospore production in a liquid culture and the virulence of blastospores compared to conidia towards *Spodoptera frugiperda* caterpillars. We also used comparative transcriptomics to identify genes and protein families that are actively expressed during the infection process of insects in vivo to investigate why blastospores seem more virulent than conidia.

A previous study found that blastospores of different isolates of *B. bassiana* cultivated in media with glucose concentrations greater than 220 g L^−1^ were as much as 53% smaller in volume compared to blastospores produced in media with glucose concentrations of 40–100 g L^−1^ [15]. We observed a reduction in the *Metarhizium anisopliae* blastospore volume produced under osmotic stress. Fewer blastospores were also produced under osmotic stress during the liquid fermentation of *Beauveria bassiana*, which has been suggested to be an osmoadaptation attributed to the concentration of solutes in the cytoplasm [15]. In the yeast (*Saccharomyces cerevisiae*), the dynamics of osmostress-induced cell volume loss is a consequence of osmotic adaptation to restore optimal diffusion rates for biochemical and biological cell processes [36]. In contrast, we observed that *Metarhizium rileyi* isolates have more mycelia and hyphal bodies under osmotic stress conditions, which imply that it is not possible to make generalizations for all entomopathogenic fungi studied to date.

Changes in nutrition, pH, and temperature can affect fungal morphogenesis and the phenotype of fungal cells [37]. The dimorphism between yeast-like growth and mycelial formation in the entomopathogenic fungus *M. rileyi* under in vitro conditions is impacted by specific nutritional conditions [38]. During the in vivo replication of *M. rileyi*, a tightly orchestrated development program involving specific chemical quorum-sensing chemicals mediate the transition from hyphal bodies to mycelia [38]. Such a correlated switch in the fungal cell phenotype is likely correlated to cell population density and indicates the production of autoinducers mediating controls in the developmental programs that modulate either cell behavior or phenotype [39].

Entomopathogenic fungi are very heterogeneous, and their growth patterns differ extensively between species. The *Beauveria bassiana* [8,15] and *M. anisopliae* isolates studied here supported our hypothesis of higher blastospore production under high osmotic stress conditions, but in contrast the *M. rileyi* isolates did not. More studies are needed to understand the role of osmotic stress in inducing yeast-like growth and in the formation of blastospores in liquid cultures of *M. rileyi*.

The germination and penetration time point of *M. anisopliae* and *M. rileyi* propagules were also studied here. Our results are in concordance with those of previous studies [7,40,41], and the time to germination, appressoria formation, and penetration found in our SEM analysis revealed that blastospores germinate and penetrate between 2 and 9 h. In contrast, conidia germinate and penetrate between 16 and 32 h after spraying.

Entomopathogenic fungi infect susceptible hosts via direct cuticle penetration, with the initial contact between the fungal propagule and the insect epicuticle potentially determining the outcome of the interaction [42]. Thus, the cuticle is the first and most significant barrier to entomopathogenic fungi and an important immune component in insects. Dynamic interactions at the cuticle surface influence the pathogens’ ability to infect hosts and entomopathogenic fungi’s evolved mechanisms for the adhesion and recognition of the host surface. These specific adaptations include the production of specialized infectious cellular structures (e.g., appressoria or penetrant tubes); hydrolytic, assimilatory, and/or detoxifying enzymes (e.g., lipase/esterases, catalases, cytochrome P450s, proteases, and chitinases); and secondary and other metabolites that facilitate infection [30].

Delayed penetration can prolong the exposure of fungal propagules to biotic and abiotic factors that are harmful to the pathogen [43]. Furthermore, delaying entry to the hemocoel allows the host to mobilize its cellular and humoral defenses [44]. Thus, successful propagules must not only be able to cope with the physical environment on the host surface but also perform against insect innate immune defenses [44]. The most external surface of the insect cuticle, the epicuticle, is formed by a thin lipid layer making it hydrophobic, which facilitates the attachment of the hydrophobic conidia [45]. However, many studies have shown that the hydrophilic blastospores can germinate, penetrate, and kill insects faster than can conidia [7,41,46].

Despite the fact that they won the battle against the insect cuticle by germinating faster than conidia, we did not observe that blastospores are more virulent compared to conidia towards *S. frugiperda* for *M. anisopliae* and *M. rileyi*. Comparing the genes expressed during *M. anisopliae* and *M. rileyi* propagule penetration and their virulence for *S. frugiperda* caterpillars, the higher mortality caused by the conidia of both fungi correlate with the high number of virulence-related protein family domains (PFAMs) expressed. Interestingly, we only found one shared PFAM between blastospores of *M. anisopliae* and *M. rileyi*. The only PFAM shared between blastospores and conidia of both fungi is cytochrome P450, an oxidative stress factor. During propagule attachment to the cuticle, entomopathogenic fungi must cope with harmful environmental conditions such as solar radiation, fluctuating humidity and temperature, antagonistic microorganisms, and antifungal compounds released by insects [42]. Thus, the infectious conidia have evolved various mechanisms to cope with these biotic and abiotic stressors, often involving genes linked with virulence and oxidative stress [42,43].

Our finding that the total gene expression profiles of conidia and blastospores differ fundamentally may be linked to a difference in the past selection pressures of these two propagules, being adapted to the external environment and proliferation to the internal body cavity of insects, respectively. Differences in transcriptomic profiles of conidia and blastospores during cuticle infection also highlight that despite both propagules being infectious, different mechanisms underlie their respective infection mechanisms, which in turn likely explains the faster germination rate of blastospores.

Selection for virulence in entomopathogenic fungi is likely governed by two opposing pressures being exerted: (a) specialization towards specific insect host species or populations, or (b) the maintainance of a broad host range increasing the number of available hosts. Here, we studied two species that seem to act differently in the environment: *M. anisopliae*, widely known for its broad host range, and *M. rileyi*, a specialist for Lepidoptera species, especially Noctuid. The virulence tests performed here shows that specialization results in greater virulence to insects, as demonstrated by the higher virulence of *M. rileyi* towards *S. frugiperda* caterpillars. However, comparative transcriptomics during propagule penetration on insect cuticles showed that this confirmed virulence may be not directly linked to a greater number of virulence-related expressed PFAM terms. In general, *M. anisopliae* propagules were less virulent towards *S. frugiperda* caterpillars, but expressed more virulence-related PFAM terms compared to *M. rileyi*. Fungal adaptive responses may explain these results as they can be mediated by epigenetic mechanisms, allowing short-term specialization while maintaining the broad host range potential [42,47]. Evidence increasingly suggests that a significant factor driving the co-evolutionary arms race between the pathogen and the host occurs on the cuticular surface—also involving the mediation of the insect cuticle microbiome [48,49].

Although considerable progress has been made in recent years, many of the molecular determinants mediating these interactions in the pathogen and the host remain uncovered [42,47]. Studies comparing broad-host-range fungi such as *M. anisopliae* with more specialist fungi such as *M. rileyi* should be particularly useful for delineating the factors involving the infection of different fungal propagules and the effective host range of entomopathogenic fungi.

## 5. Conclusions

The virulence tests performed here shows that *M. rileyi*, a Lepidoptera specialist fungi, was more virulent to *S. frugiperda* caterpillars compared to *M. anisopliae*, a generalist one. Comparative transcriptomics during propagule penetration of insect cuticles showed that, in general, *M. anisopliae* expresses more virulence and oxidative stress factors compared to *M. rileyi*. However, analyzing differences between propagules, *M. rileyi* blastospores express more virulence-related genes towards *S. frugiperda* than do *M. anisopliae*. In contrast, conidia of both fungi express more virulence-related oxidative stress factors than do blastospores. Our results highlight that blastospores use a different virulence mechanism than do conidia and increase our understanding of the fungal process of insect pathogenicity. In the future, we potentially may find gene candidates whose manipulation could lead to more effective insect biological control agents that may be explored in new biological control strategies

## Figures and Tables

**Figure 1 microorganisms-11-01594-f001:**
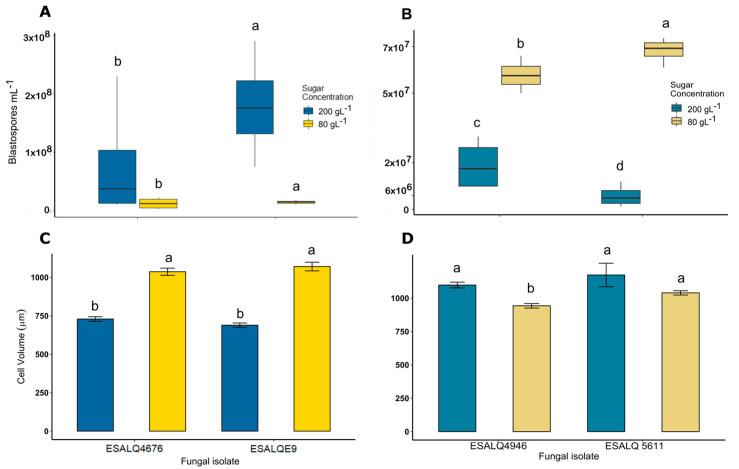
Blastospore production of *M. anisopliae* and *M. rileyi* in liquid fermentation under different levels of carbon source (80 g L^−1^ and 200 g L^−1^). (**A**) Blastospore production (yield per mL^−1^) of two isolates each of *M. anisopliae*. (**B**) Cell volume (µ3) of two isolates each of *M. anisopliae*. (**C**) Blastospore production (yield per mL^−1^) of two isolates each of *M. rileyi*. (**D**) Cell volume (µ3) of two isolates each of *M. rileyi.* Experiments were independently repeated three times. Means (±SD) followed by non-corresponding letters are significantly different (Tukey’s test, *p* < 0.05).

**Figure 2 microorganisms-11-01594-f002:**
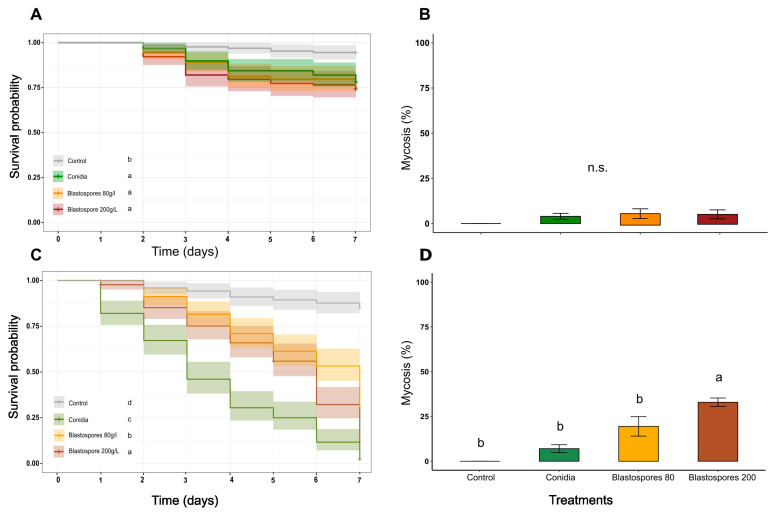
Survival probability and the percentage of mycosed cadavers of *Spodoptera frugiperda* caterpillars treated with *Metarhizium* spp. conidia and blastospores produced in two sugar source concentrations (80 and 200 g/L). (**A**) Survival of *S. frugiperda* caterpillars exposed to conidia and blastospores of *M. anisopliae* for 7 days (**B**) Percentage of *S. frugiperda* cadavers colonized by *M. anisopliae*. (**C**) Survival of *S. frugiperda* caterpillars exposed to conidia and blastospores of *M. rileyi* for 7 days (**D**) Percentage of *S. frugiperda* cadavers colonized by *M. anisopliae*. The letters in front of the captions and above the bars indicate the statistical difference between the treatments; equal letters represent treatments that did not differ from each other, while different letters indicate a statistical difference at a 0.05% probability level.

**Figure 3 microorganisms-11-01594-f003:**
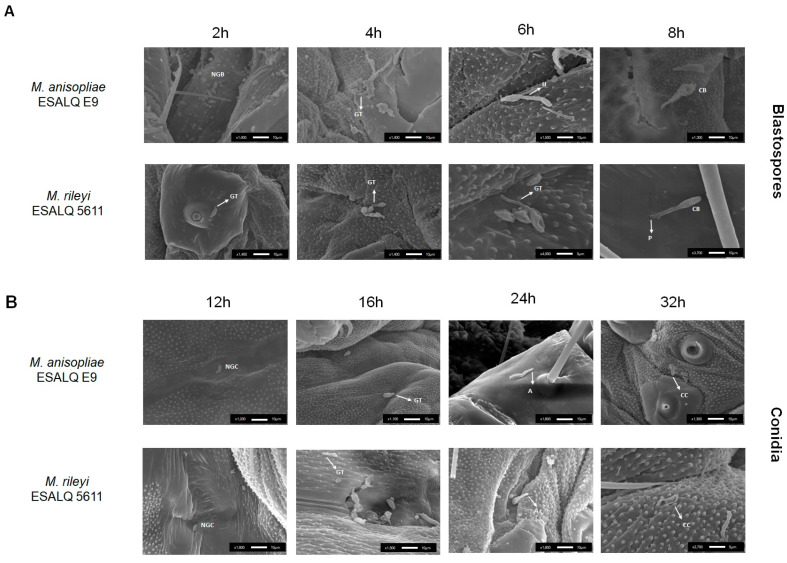
Scanning electron microscopy micrographs of *Metarhizium* spp. propagules during germination on *Spodoptera frugiperda* integument for blastospores. Briefly, 2, 4, 6, and 8 h post-exposure for blastospores (**A**), and 12, 16, 24, and 32 h post-exposure for conidia (**B**). NGB: Non-geminated blastospores; NGC: Non-germinated conidia; GT: germ tube; H: hyphae; CB: collapsed blastospore post-infection; P: propagule penetration; A: appressoria; CC: collapsed conidia post-infection.

**Figure 4 microorganisms-11-01594-f004:**
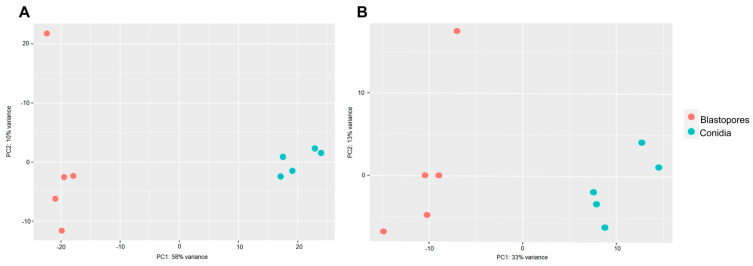
Principal component analysis of regularized logarithmic (rlog)-transformed gene counts of *Metarhizium anisopliae* and *M. rileyi* propagules during penetration through *Spodoptera frugiperda* caterpillars. (**A**) *M. anisopliae* samples. (**B**) *M. rileyi* samples. Orange and green dots represent blastospore and conidium samples, respectively.

**Figure 5 microorganisms-11-01594-f005:**
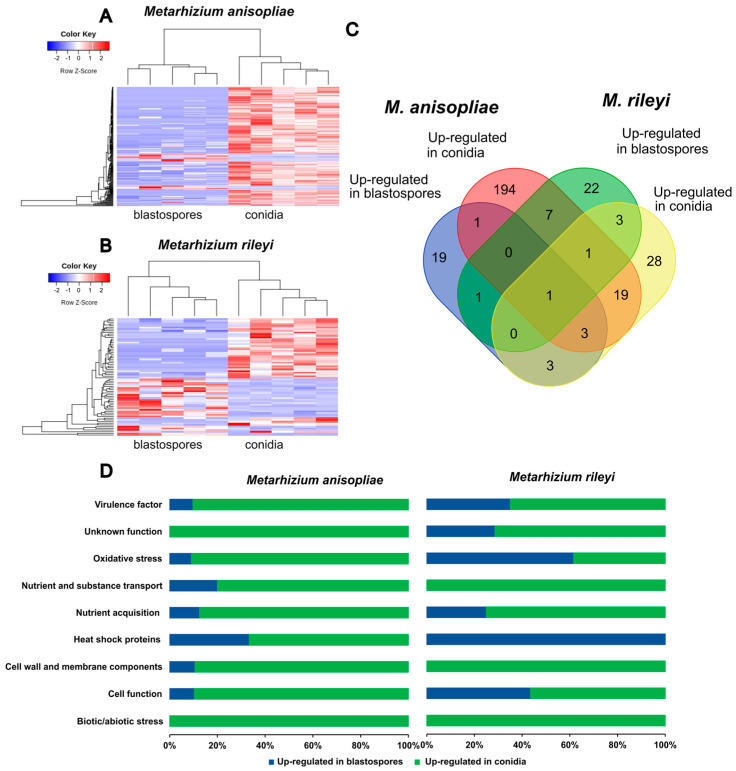
Differential gene expression between blastospores and conidia on *Spodptera frugiperda*. (**A**) Heatmap of the 319 differentially expressed genes (Padj < 0.1) in *Metarhizium anisopliae* blastospores compared to those in conidia during penetration of *S. caterpillars*. (**B**) Heatmap of the 101 differentially expressed genes (Padj < 0.1) in *Metarhizium rileyi* blastospores compared to those in conidia during penetration of *S. frugiperda* caterpillars. (**C**) Venn diagram of up-regulated PFAM terms expressed in *M. anisopliae* and *M. rileyi* propagules during penetration of *S. frugiperda* cuticle. (**D**) Percentage of PFAM terms up-regulated in *M. anisopliae* and *M. rileyi* propagules during penetration of *S. frugiperda* caterpillar’s cuticle (Log2FC < 2; *p* < 0.001).

**Figure 6 microorganisms-11-01594-f006:**
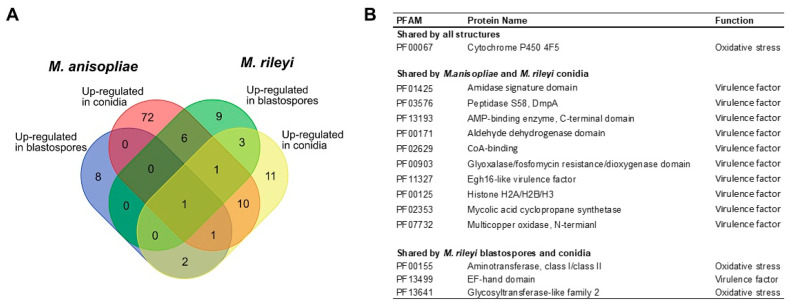
(**A**) Venn diagram of virulence-related up-regulated protein families expressed in *Metahrizium anisopliae* and *Metarhizium rileyi* propagules during penetration of *Spodoptera frugiperda* cuticle. (**B**) Up-regulated protein families related to virulence factors and oxidative stress found in *M. anisopliae* and *M. rileyi* propagules during penetration of *S. frugiperda* cuticle.

**Table 1 microorganisms-11-01594-t001:** Estimated median lethal time (LT50), 95% lethal time (LT95), lower confidence interval (lci) and upper confidence interval (uci) of the *Spodoptera frugiperda* caterpillars after being sprayed with *Metarhizium anisopliae* propagules. Time is measured in days.

	Letal Time 50 (LT50)			Letal Time 95 (LT95)		
Fungal Propagule	Estimated	lci *	uci *	Estimated	lci	uci
Conidia	17.5	11.6	47.8	162	55.8	2306
Blastospores 80 g L^−1^	18.5	11.8	58.3	245	71.1	6394
Blastospores 200 g L^−1^	20.6	12.1	99.4	530	106	71405

* lci = 95% lower confidence interval and uci = 95% upper confidence interval.

**Table 2 microorganisms-11-01594-t002:** Estimated median lethal time (LT50), 95% lethal time (LT95), lower confidence interval (lci) and upper confidence interval (uci) of the *Spodoptera frugiperda* caterpillars after being sprayed with *Metarhizium rileyi* propagules. Time is measured in days.

	Letal Time 50 (LT50)			Letal Time 95 (LT95)		
Fungal Propagule	Estimated	lci *	uci *	Estimated	lci	uci
Conidia	2.57	1.97	3.12	10.2	7.23	20.2
Blastospores 80 g L^−1^	6.28	5.84	6.88	18.4	15.3	27.3
Blastospores 200 g L^−1^	4.84	3.99	6.12	13.3	9.06	39.4

* lci = 95% lower confidence interval and uci = 95% upper confidence interval.

## Data Availability

The data presented in this study are openly available in the National Library of Medicine (NCBI), BioProject accession number PRJNA976674.

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
