# Peer review of "Blastospores from Metarhizium anisopliae and Metarhizium rileyi Are Not Always as Virulent as Conidia Are towards Spodoptera frugiperda Caterpillars and Use Different Infection Mechanisms"

_microorganisms, 2023, doi:10.3390/microorganisms11061594_

Round 1
Reviewer 1 Report
he authors compared the different virulence of Metarhizium anisopliae and M. rileyi blastopores and conidia towards the caterpillar pest Spodoptera frugiperda. And further comparative transcriptomics analysis show that there were different virulence-related genes expression in the two species and in the blastopores and conidia propagules. Their findings are important for the delineating factors involving infection of different fungal propagules and the effective host range of entomopathogenic fungi. I provided some comments for the authors to consider as outlined below.
General comments:
1. Line 133-136 More comprehensive culture information of fungal conidia production should be present, especially in the M. rileyi.
2. Line 256-258 It is different blastopores yield in different carbon source concentration. As we all known, nitrogen source also is the factor that effect blastopores and conidia yield. Therefore, it is better to show the investigation on blastopores and conidia yield in different type of carbon and nitrogen source.
3. Line 265 Please give the definition of µ3 in the Material and Methods.
4. More information of these PCA data in Fig. 4A and Fig. 4B should provide.
5. The describe should be characterized by logicality, systematization and succinctness in the DISCUSSION. Therefore, the part should be revised.
Other comments:
1. Line 139: The “ANOVA” should present the full name.
2. Line 222-225 LEADING:20 … should note the information of reference.
3. Line 234 The accession codes should provide.
4. References should be revised carefully.
Author Response
Dear reviewer,
Thank you for revising and improving this manuscript with your comments.
We tried to make the adjustments according to each topic that was mentioned in the report. You can find our reply attached.
The only thing we still need to do is provide the accession codes for the samples. We´re still waiting for the NCBI upload, but we´ll send it as soon as possible.
Please, let us know in case of any doubts or any other necessary adjustments.
Best Regards,
Isabella, Camila, Henrik, and Italo.

Reviewer 2 Report
Microorganisms-2392941- 12 May 2023
Blastospores from Metarhizium anisopliae and Metarhizium rileyi are not always as virulent as conidia towards Spodoptera frugiperda caterpillars and use different infection mechanisms.
Comments and Suggestions for Authors
1.- Line 103: give more description of the larval diet or add a reference.
2.- Line 104: add how many days of development had the 2o. instar larvae.
3.- Lines 140-141.- add before what it means (ghlt) and (hnp), maybe “general linear hypotheses and half-normal plots, respectively.
4.- Line 142, add “fungus species” at “fungus”
5.- Line 160, add the number of CFU in suspension fungal for each treatment.
6.- Line 165-166: dead larvae were disinfected or not before placing them to humid chamber. If this was not the case, explain.
7.- Line 179-184: abbreviation should be used for the scientific name of the genera.
8.- Line 185: change Scanning Electron Microscopy by SEM.
9.- Line 206: Is correct for the scientific name of the species.
10.- Line 234: What does it mean.
11.- Line 240: add the statistical version of software R.
12.- Line 386: Improve the quality of figure 4.
13.- Line 398: Is correct the name of S. caterpillars.
14.- lines 526-611: In the reference section, the various scientific names of organisms should be in italics style.
15.- Give more explanation why using comparative transcriptomics during propagule penetration on insect cuticles M. rileyi expressed more virulence against S. frugiperda than M. anisopliae.
Author Response

(The authors gave the same response as above.)

Reviewer 3 Report
The manuscript titled “Blastospores from Metarhizium anisopliae and Metarhizium rileyi are not always as virulent as conidia towards Spodoptera frugiperda caterpillars and use different infection mechanisms” represents an interesting study devoted to comparing the virulence of conidia and blastospores of entomopathogenic fungi M. anisopliae and M. rileyi against Spodoptera frugiperda and to identifying virulence-related genes expressed specifically during cuticle penetration, using comparative transcriptome analysis.
The manuscript is well written and the results obtained support the conclusions raised up by the Authors. However, there are some minor changes (mostly technical ones) to be addressed:
Title
Remove the dot from the end of the title. Also, consider changing the title.
Introduction
Line 36: are fungi from the genus Metarhizium
Line 41: Rephrase the sentence as follows: “The species M. anisopliae (Metsch.) Sorokin, which is used to control sugarcane pests in Brazil, is one of the best examples of using an entomopathogenic fungus in pest control [5].“
Line 43: ‘in vito’ should be italic
Line 48: blastospores are naturally formed
Materials and Methods
Line 97: Replace “:” with “.”
Line 134: The authors can use abbreviations for Potato Dextrose Agar and Sabouraud Maltose with Yeast Extract since they are introduced earlier
Line 160: 2 mL
Line 176: species
Line 190: with blastospores; with aerial conidia
Line 243: Explain the abbreviation “PCA” upon the first use
Results
Line 259: two M. anisopliae isolates; responds
Line 290-292: The sentence can be deleted. It is materials and methods
Lines 328-329: Use abbreviations for Metarhizium anisopliae, Metarhizium rileyi, and Spodoptera frugiperda throughout the text
Line 327-329: The sentence can be deleted. It is materials and methods
Lines 329-330: Does the sentence refer to blastospores of both fungal species? Please specify
Line 33-336: The sentence is unclear. Please rephrase it.
Line 347-350: Sentences can be deleted. They are already written in Materials and methods section.
Line 351: were uniquely mapped to the respective fungi's genome
Line 357: PCA of the M. anisopliae and M. rileyi
Line 367: differentially
Line 369: Explain the abbreviation “PFAM” upon the first use
Discussion
Line 418: ‘in vivo’ should be italic
Line 422-423: It is sufficient to refer to the reference only at the end of the sentence
Lines 445-447: The sentence is unclear. Please rephrase it.
Author Response

(The authors gave the same response as above.)

Reviewer 4 Report
I have read the manuscript (microorganisms-2392941). Entitle: Blastospores from Metarhizium anisopliae and Metarhizium rileyi are not always as virulent as conidia towards Spodoptera frugiperda caterpillars and use different infection mechanisms written by Gotti et. al., for publication of microorganisms/MDPI. In this study, they compared the virulence of conidia and blastospores of two EPF species against S. frugiperda.
I suggest that the author add a few sentences introducing pest species, i.e. S. frugiperda in the introduction section.
Please add the suitable references: Lines 55 to 62
Remove: Lines 63 to 71 or include them in the discussion section.
Add suitable references to support the lines 81-83 (There are several findings on the natural infestation of M. rileyi in the field, for example: Identification of entomopathogenic fungus Metarhizium rileyi infested in fall armyworm in the cornfield of Korea, and evaluation of its virulence (https://doi.org/10.1002/arch.21965), High virulence of a naturally occurring entomopathogenic fungal isolate, Metarhizium (Nomuraea) rileyi, against Spodoptera frugiperda (https://doi.org/10.1111/jen.13007).
Please re-write lines 85-87.
Please provide detail information about the artificial diet which you feed for S. frugiperda, Line 103
How do you recognize the dead larvae and confirm the mortality is caused by EPF? Line 165
Why you selected only 2nd instar larvae?
Please mention the specific concentrations of EPFs for virulence test. How long did you check the larval mortality?
Please describe which statistical analysis, software and method you used for each analysis in a separate topic (Statistical analysis) in the materials and methods section.
Please use big-size figures, and use appropriate font size in figures. It is hard to read the text in figures.
Survival graph is not clear, please use high-resolution figures.
Please provide the conclusion of your study. Please provide the key messages based on your research outcomes only. I would love to read striking points and take-home messages that will linger in the readers’ minds. What is the novelty, how does the study elucidate some questions in this field and the contributions the paper may offer to the scientific community?
References: Please double-check the citations, their style, spell check, and other grammatical errors. moreover, the author should cut the old and less matching literature and include the latest literature some of them are above.
Overall after I evaluate and request the author for this manuscript as a “MAJOR REVISION”.
Good Luck!
Author Response

(The authors gave the same response as above.)

Round 2
Reviewer 4 Report
I can tell that the authors put effort to improve this manuscript. However, there have some errors which should address before it is suitable for publication.I suggested to the author to add suitable references for lines 84-86 but the author didn't provide it. Please, add references to support this sentence. Please check my previous suggestion.
Lines 91-93: Rewrite: The fall armyworm, Spodoptera frugiperda (J.E. Smith (Lepidoptera: Noctuidae), is a polyphagous pest reported to infest at least 353 host plant species. Please add a reference for this statement (https://hdl.handle.net/10520/EJC-112bc26060 Montezano et al., 2018 Host plants of Spodoptera frugiperda (Lepidoptera: Noctuidae) in the Americas)
Line 96: Virulence of .................spores against S. frugiperda larvae.
Lines: 118-120, please remove it. this is not important information.
I didn't get appropriate answer for why the author selected only the 2nd instar larvae. Did you check the virulence in other stages too or did you just follow the previous publications? Which kind of behaviour of S. frugiperda makes the biological control more efficient in the field at 2nd instar larvae?
Lines 177-179: Please mention the Petri dish size (60/90 mm or ?)
How you applied fungal suspension on larvae?
During the virulence assay, did the author provides any foods for the larvae? How 2nd instar larvae survived for 7 days without any food?
Did you check the normality of mortality data?
Please calculate the LT50 and LT95 values for each treatment. This is more important to compare the virulence of any components.
line 193: function glht not the ghlt
Please add "7 days" in X axis of Figure 2 A and C (change the days interval for x-axis)
References: Please check all the references carefully.
See Reference #25
Thank you!
Author Response
Dear editor and reviewer,
Thank you for revising and improving this manuscript with your comments.
We have made all the requested adjustments according to each point raised in the report of reviewer no. 4, and point-by-point responses are provided below.
Best Regards,
Isabella, Camila, Henrik, and Italo.
- I can tell that the authors put effort to improve this manuscript. However, there have some errors which should address before it is suitable for publication.
Thank you for your comments.
- I suggested to the author to add suitable references for lines 84-86 but the author didn't provide it. Please, add references to support this sentence. Please check my previous suggestion.
We have added multiple references to requested sentences, including one reference suggested by the reviewer. Specifically, we now refer to reference 18-21 in these sentences:
- Fronza, E.; Specht, A.; Heinzen, H.; de Barros, N.M. Metarhizium (Nomuraea) rileyi as biological control agent. Biocontrol Science and Technology 2017, 27, 1243-1264, doi:10.1080/09583157.2017.1391175.
- Grijalba, E.P.; Espinel, C.; Cuartas, P.E.; Chaparro, M.L.; Villamizar, L.F. Metarhizium rileyi biopesticide to control Spodoptera frugiperda: Stability and insecticidal activity under glasshouse conditions. Fungal Biology 2018, 122, 1069-1076, doi:10.1016/j.funbio.2018.08.010.
- Zhong, K.; Liu, Z.C.; Wang, J.L.; Liu, X.S. The entomopathogenic fungus Nomuraea rileyi impairs cellular immunity of its host Helicoverpa armigera. Archives of Insect Biochemistry and Physiology 2017, 96, 10, doi:10.1002/arch.21402.
- Yan-li, Z.; Hui, D.; Li-sheng, Z.; Zu-min, G.; Jin-cheng, Z. High virulence of a naturally occurring entomopathogenic fungal isolate, Metarhizium (Nomuraea) rileyi, against Spodoptera frugiperda. Journal of Applied Entomology 2022, 146, 659-665, doi:https://doi.org/10.1111/jen.13007.
- Lines 91-93: Rewrite: The fall armyworm, Spodoptera frugiperda (J.E. Smith (Lepidoptera: Noctuidae), is a polyphagous pest reported to infest at least 353 host plant species. Please add a reference for this statement (https://hdl.handle.net/10520/EJC-112bc26060 Montezano et al., 2018 Host plants of Spodoptera frugiperda(Lepidoptera: Noctuidae) in the Americas)
Ok. The sentence was rewritten, and the new reference [22] was updated (Line 92) so we now include the reference as suggested by the reviewer:
- Montezano, D.; Specht, A.; Sosa-Gómez, D.; Roque-Specht, V.; Sousa-Silva, J.; Paula-Moraes, S.; Peterson, J.; Hunt, T. Host Plants of Spodoptera frugiperda (Lepidoptera: Noctuidae) in the Americas. 2018.
- Line 96: Virulence of .................spores against S. frugiperda larvae.
Ok, we have added “larvae” to the sentence as suggested.
- Lines: 118-120, please remove it. this is not important information.
Ok. The sentences were removed from the manuscript.
- I didn't get appropriate answer for why the author selected only the 2nd instar larvae. Did you check the virulence in other stages too or did you just follow the previous publications? Which kind of behaviour of S. frugiperda makes the biological control more efficient in the field at 2nd instar larvae?
The virulence was only evaluated against second-instar caterpillars. Generally, this instar is used to test the virulence of fungi for two reasons: 1) 1° instar are highly susceptible and naturally experience high mortality rates (Islam et al., 2023); 2) in practical terms, caterpillars larger than the third instar are less susceptible and have a habit of seeking refuge in the whorls of corn plants, where the probability of fungal infection decreases (Pannuti et al., 2016)
This additional info has now been included in the manuscript (Lines 183-187), and the above-mentioned references included as reference [27, 28].
References:
- ISLAM, S. M. N. et al. Biocontrol potential of native isolates of Beauveria bassiana against cotton leafworm Spodoptera litura (Fabricius). Scientific Reports, v. 13, n. 1, 2023-05-23 2023. ISSN 2045-2322. Accessed on: 2023-05-26T12:49:57.
- PANNUTI, L. E. R. et al. On-Plant Larval Movement and Feeding Behavior of Fall Armyworm (Lepidoptera: Noctuidae) on Reproductive Corn Stages. Environmental Entomology, v. 45, n. 1, p. 192-200, 2016-02-01 2016. ISSN 0046-225X. Available at: < https://digitalcommons.unl.edu/cgi/viewcontent.cgi?article=1436&context=entomologyfacpub >. Accessed on: 2023-05-26T12:58:32.
- Lines 177-179: Please mention the Petri dish size (60/90 mm or ?)
We used 90 mm Petri dishes. The size was added to the manuscript on line 177.
- How you applied fungal suspension on larvae?
We used a Potter Spray Tower (Burkard Manufacturing Co Ltd). The apparatus contains a spray nozzle and generate aerosol particles from the fungal suspension. The information is provided at line 172-175.
- During the virulence assay, did the author provides any foods for the larvae? How 2nd instar larvae survived for 7 days without any food?
The caterpillars were fed on corn leaves during the evaluation period. The information was clarified in line 178.
- Did you check the normality of mortality data?
We did not compare the absolute mortality data, only the survival curves. Survival data do not follow a normal distribution and it is not necessary to check for normality.
- Please calculate the LT50 and LT95 values for each treatment. This is more important to compare the virulence of any components.
Ok, we have added the LT50 and the LT95 to the text.
Lines 195-196 on the methods
Lines 317-319 on the results for M. anisopliae
Lines 322-326 on the results for M. rileyi.
We also have added a table showing all the results: Tables 1 and 2.
- line 193: function glht not the ghlt
We apologize for the error. The correct name of the function is "glht." The information has been corrected in the manuscript at line 193.
- Please add "7 days" in X axis of Figure 2 A and C (change the days interval for x-axis)
The x-axis now reads “Time (days)” and the markings show the axis goes from 0 to 7 days. To make it clearer, we added “for 7 days” on Figure A and C legend.
- References: Please check all the references carefully.
Ok. All references were carefully checked.
- See Reference #25
Ok. Thank you. We have carefully edited all references, including this reference 25.
